# DreamLCM: Towards High Quality Text-to-3D Generation Via Latent Consistency Model

Submission Id: 1610

## ABSTRACT

Recently, the text-to-3D task has developed rapidly due to the appearance of the SDS method. However, the SDS method always generates 3D objects with poor quality due to the over-smooth issue. This issue is attributed to two factors: 1) the DDPM single-step inference produces poor guidance gradients; 2) the randomness from the input noises and timesteps averages the details of the 3D contents. In this paper, to address the issue, we propose DreamLCM which incorporates the Latent Consistency Model (LCM). DreamLCM leverages the powerful image generation capabilities inherent in LCM, enabling generating consistent and high-quality guidance, i.e., predicted noises or images. Powered by the improved guidance, the proposed method can provide accurate and detailed gradients to optimize the target 3D models. In addition, we propose two strategies to enhance the generation quality further. Firstly, we propose a guidance calibration strategy, utilizing Euler solver to calibrate the guidance distribution to accelerate 3D models to converge. Secondly, we propose a dual timestep strategy, increasing the consistency of guidance and optimizing 3D models from geometry to appearance in DreamLCM. Experiments show that DreamLCM achieves state-of-the-art results in both generation quality and training efficiency.

## CCS CONCEPTS

• **Information systems → Multimedia content creation**.

## KEYWORDS

Text-to-3D Generation, Diffusion Model, Gaussian Splatting, Latent Consistency Model

## 1 INTRODUCTION

Recent advancements in Diffusion Models (DMs) [34, 35, 37] have made progress in satisfying the needs of synthesizing high-quality images given text descriptions. Besides, by training on large-scale image datasets [38] where images are coupled with detailed texts, DMs achieve powerful ability in generating all kinds of 3D contents conditioned on the given text prompts. Existing works [3, 17, 22, 23, 27, 32, 44, 47] have been proposed to apply well-trained diffusion models to the task of text-to-3D generation. Conditioned on text prompts, DMs can generate guidance information in the latent

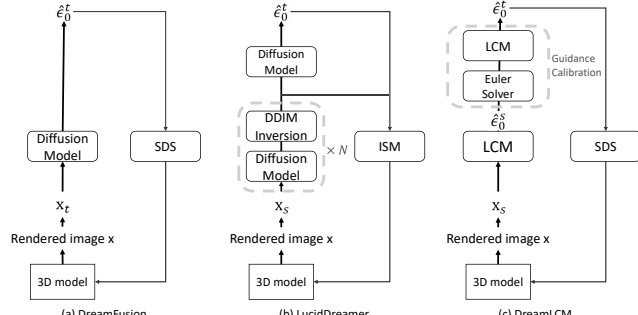

**Figure 1: Illustration of different guidance generation approaches.** $x$ and $\hat{\epsilon}$ indicates the rendered image and the guidance, respectively. (a) SDS generates guidance via a single diffusion model while producing over-smooth results. (b) LucidDreamer utilizes the DDIM inversion technique, forwarding Diffusion Models multiple times where $N = \{2, 3, 4, 5\}$. (c) The proposed DreamLCM method incorporates LCM as the guidance model. We also propose a guidance calibration strategy that uses Euler Solver to refine the guidance $\hat{\epsilon}_0^s$ to $\hat{\epsilon}_0^t$. Our method generates higher-quality guidance compared to (a) and (b).

space. This latent guidance can be utilized to supervise the carving process of the target 3D objects. Thus, this alternative approach tackles the challenge of 3D object generation without large-scale 3D models for training. For example, the Score Distillation Sampling (SDS) objective is introduced in DreamFusion [32] to leverage the robust prior knowledge acquired by text-to-image diffusion models [35, 37]. The SDS backpropagates the gradients from the 2D diffusion model to 3D objects and bridges the gap between the diffusion models and the 3D representations, as shown in Fig. 1(a). The acquired prior knowledge is utilized to optimize the 3D objects represented by Neural Radiance Fields (NeRF) [28] conditioned on a single text prompt.

However, SDS is limited in generating fine details as it produces over-smooth results. This effect has been noted by previous works [17, 22, 45]. These works attempt to improve SDS and achieve good results in increasing the quality of 3D models. For instance, ProlificDreamer [45] optimizes multiple 3D models simultaneously. These models are mutually benefited and merged by finetuning a LoRA model [12]. The LoRA thus reserves the details of the 3D model. Nevertheless, extra resources are needed to regenerate the lost details. In this paper, we think that the problem of the over-smooth issue stems from two factors. Firstly, the guidance of SDS is derived by the DDPM [9] single-step inference, which leads to *low-quality guidance* and blurred details. Secondly, the rendered images of the target 3D object act as conditions and are fed into DMs

after adding random noises. Besides, DMs need to sample timesteps randomly for the diffusion process. The randomness from both the added noises and timesteps directly results in the *inconsistent guidance* between different iterations. This inconsistency ultimately averages the details of 3D models and leads to the over-smooth issue.

This paper endeavors to tackle the over-smooth issue by handling the factors above accordingly. We propose a novel approach to generate clear guidance for the *low-quality guidance issue*. Inspired by Latent Consistency Model (LCM) [26], we propose **DreamLCM** by incorporating LCM as a guidance model to fully utilize the capability to generate high-quality guidance in a single-step inference. Notably, LCM generates high-quality images in a single-step inference, rather than gradually approaching to the origin along the probably flow ODE (PF-ODE) trajectory [42] by multi-step inference like DDIM [40]. Therefore, DreamLCM merely predicts guidance of a rendered image by directly denoising the noisy latent from an arbitrary timestep along the PF-ODE trajectory to keep fine details of the target 3D models. For the *inconsistency issue*, we observe that LCM generates consistent guidance regardless of the randomness. To solve the issue, a similar method has been proposed in LucidDreamer [22], which uses the DDIM inversion technique [40] to improve the consistency of the guidance. However, different from LucidDreamer, the proposed DreamLCM method provides two merits: 1) DreamLCM only needs a single-step inference to compute the guidance while LucidDreamer forwards the U-Net [36] in DM multiple times; 2) DreamLCM keeps the original SDS loss. Since LCM can resolve the two issues causing the over-smooth issue, there is no need to change the loss forms. On the contrary, LucidDreamer utilizes a complex objective function to adapt the DDIM Inversion method. The difference is shown in Fig. 1(b)(c).

In addition, to further improve generation quality, we propose two novel strategies, i.e., *Guidance Galibration* and *Dual Timestep Strategy*. For *Guidance Calibration*, we propose a two-stage strategy that repeats the perturbing and denoising steps to calibrate the distribution of the guidance. In this way, the disturbing information can be gradually removed. In the first stage, we perturb a rendered image and predict the corresponding guidance. This guidance is consistent with the rendered image, i.e., the 3D object, as the added noise and timestep are small. In the second stage, we run a discretization step of a numerical ODE solver, where we use the Euler Solver to obtain a latent with relatively large noises. The large noises and timestep can provide a more reasonable optimization direction. Consequently, the calibrated guidance is ensured to be consistent with both the rendered image and the highest data density conditioned on the text prompt, effectively improving the guidance's quality. We calculate SDS loss and back-propagate the gradient using the calibrated guidance to optimize the 3D models. For *Dual Timestep Strategy*, we utilize the timestep sampling strategy to enable dreamLCM to optimize the geometry and appearance of 3D objects in separate phases. In particular, in the initial phase, we apply large timesteps to guide the 3D model in producing large deformations. In this case, DreamLCM tends to optimize geometry, where the position of Gaussian Splatting is greatly updated. In the refinement phase, we use small timesteps to optimize the appearance because small timesteps help DreamLCM generate guidance with fine details. Besides, we sample monotonically decreasing

timesteps to increase the consistency of the guidance. Overall, our proposed dual timestep strategy is the combination of the timestep strategy in HiFA [49] and ProlificDreamer [45].

We apply the Gaussian Splatting [18] as the 3D representation to form the 3D target objects. The proposed DreamLCM achieves the state-of-the-art results. As shown in Fig. 3, we can see that DreamLCM generates high-quality 3D objects with fine details. Besides, our model trains end-to-end, reducing training costs and maintaining a streamlined training pipeline. Overall, our contributions can be summarized as follows:

- We resolve the over-smooth issue of SDS in a new perspective by proposing DreamLCM. We analyze the two weaknesses in the generated guidance of diffusion models, i.e., low quality and low consistency. In response to the two issues, we incorporate LCM as our guidance model to make full use of the ability in LCM and generate high-quality, consistent guidance in a single inference step.
- We propose two novel strategies to further improve the quality of the guidance for 3D generation. A guidance calibration strategy is proposed, using Euler solver to obtain an improved sample, which subsequently generates calibrated guidance to help 3D models converge accurately. Besides, a dual timestep strategy is proposed, enabling DreamLCM to optimize the geometry and the appearance in two phases. We prove the effectiveness of the two strategies in Sec.6.4.
- We conduct experiments to demonstrate that DreamLCM significantly outperforms the state-of-the-art methods in terms of both quality and training efficiency.

## 2 RELATED WORK

### 2.1 Diffusion Models

Diffusion Models(DMs) have emerged as powerful tools for image generation [10, 29, 31, 34, 40, 42], excelling in denoising noise-corrupted data and estimating data distribution scores. The stable ability of DMs for generating high-quality images led to multiple applications in various domains, including video [8, 11, 19] and 3D [32, 44], *etc*. During inference, these models employ reverse diffusion processes to gradually denoise data points and generate samples. In comparison to Variational Autoencoders (VAEs) [20, 39] and Generative Adversarial Networks (GANs) [7], diffusion models offer enhanced training stability and likelihood estimation. However, their sampling efficiency is often hindered. Discretizing reverse-time SDE [6, 42] or ODE [42] are proposed to handle the challenge, various techniques such as ODE solvers [24, 25, 40, 48], adaptive step size solvers [14], and predictor-corrector methods [42] have been proposed. Notably, the Latent Diffusion Model(LDM) [35] conducts forward and reverse diffusion processes in the latent data space, leading to more efficient computation. The Consistency model [26, 41] demonstrates promising potential as a rapid sampling generative model for generating high-quality images in a single-step inference. In this paper, we transfer the ability of LCM to text-to-3D task to generate high-quality guidance. Meanwhile, we use Euler Solver as the numerical ODE Solver to further calibrate the guidance for higher quality.

## 2.2 Text-to-3D Generation.

This task targets generating 3D contents from given text prompts. The 3D contents are parameterized by various 3D representations, including implicit representations [1, 2, 5, 16], 3D Gaussians [4, 21, 22, 43, 46], *etc*. Existing methods includes 3D generative methods [15, 30]. However, these methods can only generate objects within limited categories due to the lack of large-scale datasets. Our method uses DMs to guide the 3D optimization. DreamField [13] represent pioneering efforts in training Neural Radiance Fields (NeRF) [28] with guidance from CLIP [33]. Dreamfusion [32] firstly employs Score Distillation Sampling (SDS) to distil 3D assets from pretrained text-to-image diffusion models. SDS has become integral to subsequent works, with endeavours aiming at enhancing 3D representations, addressing inherent challenges such as the Janus problem, and mitigating the over-smooth effect observed in SDS. Recent studies like ProlificDreamer [45], HiFA [49], and LucidDreamer [22] have made significant strides in improving the SDS loss. Concurrent methods such as CSD [47] and NFSD [17] provide empirical solutions to enhance SDS. In our novel approach, DreamLCM, we resolve the over-smooth problem in a new perspective of the guidance model, showing that it is possible to generate high-quality 3D models without any alterations to SDS.

## 3 REVISITING CONSISTENCY MODELS

The core idea of the Consistency Model (CM) and Latent Consistency Model (LCM) is to learn a function that maps any points on a trajectory of PF-ODE [42] to that trajectory origin, i.e., the solution of PF-ODE. The trajectory origin indicates the real data distribution region, which has the highest data density. Besides, LCM extends the denoising process to the latent space. LCM predicts the solution of PF-ODE by introducing a consistency function in a single-step inference. LCM is a text-to-image DM $f_\phi$ parameterized by $\phi$. The objective is to fulfill the mapping: $f_\phi(x_t, y, t) \rightarrow x_0$, where $x_t$ is the noisy latent while $y$ is the text prompt. The self-consistency property of LCM is expressed in Eq. (1) as

$$f_\phi(x_t, y, t) = f_\phi(x_{t'}, y, t'), \forall t, t' \in [\delta, T], \quad (1)$$

where $\delta$ is a fixed small positive number. The formula shows the consistency of the perturbed images between different timesteps.

Overall, LCM has two benefits: 1) generating $f_\phi(x_t, y, t)$ with high quality in a single-step inference. 2) different $x_t$ between different $t$ generate consistent guidance. We attribute the over-smooth issue in SDS loss to two factors in Sec. 1. The first is the *low-quality guidance* issue, which can be resolved by utilizing LCM to generate high-quality guidance. The second is the *inconsistency issue*. The issue is mitigated because the guidance generated via LCM is consistent between different timesteps. Therefore, we incorporate LCM into the text-to-3D task as the guidance model. The guidance generated by LCM exhibits high quality and high consistency.

## 4 METHOD

In this section, we present DreamLCM for high-quality text-to-3D synthesis. First, we formulate the entire 3D generation process and analyze the issues in recent works. Then, we propose DreamLCM, Guidance Calibration, and Dual Timestep Strategy. We explain how these methods benefit the generation quality.

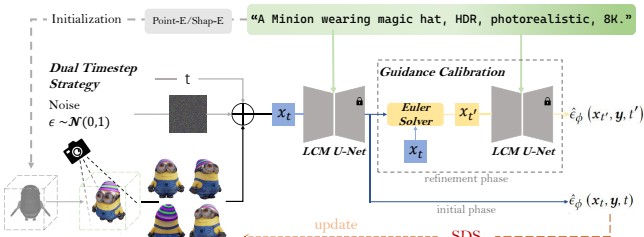

**Figure 2: Illustration of DreamLCM. DreamLCM initializes the 3D model $\theta$ via text-to-3D generator [15, 30]. We utilize the proposed timestep strategy to divide the training into two phases. In the initial phase, we directly generate guidance via a single LCM network. In the refinement phase, we utilize another LCM network and an Euler Solver to calibrate the guidance. We calculate the original SDS loss to update $\theta$.**

## 4.1 The Problem Definition

Dreamfusion [32] proposes a general framework for the text-to-3D generation task. It has two important components. The first is a 3D representation, e.g., NeRF [28], 3D Gaussian Splatting [18], that is parameterized by $\theta$ for depicting the target 3D object $\Theta$. The second is a pretrained text-to-image diffusion model for providing guidance information and supervising the target $\Theta$. To bridge the 3D object and its guidance model, a differentiable renderer $g$ is utilized to obtain the rendered image $x$, which is formulated as $x = g(\theta, c)$ with camera pose $c$. Then $x$ is fed into a VAE encoder [20] and perturbed by noise $\epsilon$. Here, we denote the latent embedding as $x$ for simplification. Given noisy latent $x_t$, timestep $t$, and text prompt $y$ as inputs, the guidance model predicts guidance gradients for updating the 3D object. The guidance prediction of DM can be expressed by two equivalent forms, i.e., $\epsilon$-prediction $\hat{\epsilon}_\phi(x_t, y, t)$ and $x$-prediction $\hat{x}_0^t$ in Eq (2) following SDS [32].

$$\nabla_\theta \mathcal{L}_{\text{SDS}}(\phi, g) = \mathbb{E}_{t,\epsilon,c} \left[ \omega(t) \left( \hat{\epsilon}_\phi(x_t, y, t) - \epsilon \right) \frac{\partial g(\theta, c)}{\partial \theta} \right],$$
$$= \mathbb{E}_{t,\epsilon,c} \left[ \frac{\omega(t)}{\gamma(t)} \left( x_0 - \hat{x}_0^t \right) \frac{\partial g(\theta, c)}{\partial \theta} \right]. \quad (2)$$

Here $\hat{x}_0^t = \frac{x_t - \sqrt{1-\bar{\alpha}_t}\hat{\epsilon}_\phi(x_t, y, t)}{\sqrt{\bar{\alpha}_t}}$, $\sqrt{\bar{\alpha}_t}$ is the noise weight, which shows that $\hat{\epsilon}_\phi(x, y, t)$ and $\hat{x}_0^t$ are equivalent, and we consider them both as guidance. $\omega(t)$ is a weighting function that depends on the timestep $t$. $\gamma(t) = \frac{\sqrt{1-\bar{\alpha}_t}}{\sqrt{\bar{\alpha}_t}}$. The gradient leads the 3D model closer to the corresponding text prompt.

Previous works ProlificDreamer [45] and LucidDreamer [22] observe that SDS generates over-smooth 3D models. We attribute this issue to two factors: 1) DMs [35, 37] generate low quality guidance because $\hat{x}_0^t$ are obtained from DDPM single-step inference [9], as shown in Fig. 1(a); 2) DMs are sensitive to the randomness in noise $\epsilon$ and timestep $t$. Especially, a large $t$ is hard to generate $\hat{x}_0^t$ consistent with $x_0$, which averages the appearance of the 3D models during optimization. Overall, the weakness in DMs generates poor guidance, resulting in over-smooth outcomes.

## 4.2 DreamLCM

The proposed DreamLCM method aims at resolving the aforementioned over-smooth issue by incorporating LCM and further enhancing the generation quality by proposing two effective strategies, i.e., Guidance Calibration and a dual timestep strategy. The overall framework is shown in Fig. 2. The entire DreamLCM approach is depicted in Algorithm 1.

For the *low-quality guidance issue*, DreamLCM utilizes the powerful ability of LCM to generate high-quality guidance. LCM trains a function $f_\phi$ in Eq. 1, which can be seen as guidance, to map any $x_t$ to its PF-ODE trajectory origin, i.e.. Consequently, LCM is capable of generating high-quality guidance in a single-step inference.

For the *inconsistency issue*, we utilize the important property of LCM in Eq. 1, highlighting the consistency of guidance between different timesteps. When guided by DMs, rendered image $x_0$ is added random noise $\epsilon$, which is further weighted by different timesteps $t$. The randomness in $\epsilon$ and $t$ directly results in inconsistent $\hat{x}_0^t$, eventually resulting in a *feature-average* outcome. However, due to LCM's property, LCM can generate consistent $\hat{x}_0^t$ regardless of the randomness.

Overall, given a timestep $s$, we integrate LCM and SDS by generating guidance $\hat{\epsilon}_\phi(x_s, y, s)$ via LCM to calculate the the SDS loss:

$$\nabla_\theta \mathcal{L}_{\text{SDS}}(\phi, g) = \mathbb{E}_{s,\epsilon,c}\left[w(s)\left(\hat{\epsilon}_\phi(x_s, y, s) - \epsilon\right)\frac{\partial g(\theta, c)}{\partial \theta}\right], \quad (3)$$

where $\hat{\epsilon}_\phi(x_s, y, s)$ is obtained in a single-step inference with high quality and fine details. It enables the 3D model to have fine details, mitigate the over-smooth issue, and save training costs.

Unfortunately, it is difficult to resolve the two issues perfectly due to the nature of diffusion models. The images generated by LCM with a single-step inference are always blurred, and the high-quality images are derived from four-step inferences iteratively. This fact is also stated in LCM [26]. For the first issue, if we directly utilize LCM's single-step inference results, the guidance would be blurred and not conducive to generating high-quality 3D models. Therefore, we further resolve this weakness by proposing a *guidance calibration* strategy. For the second issue, during the training of LCM, the two noisy latents in Eq. (1) are limited to be on the same PF-ODE trajectory, rather than two arbitrary noisy latents. We follow this protocol during the inference by fixing the noise $\epsilon'$ to perturb the rendered image, reducing the randomness in noise. Besides, we propose a decreasing timestep strategy, where we utilize monotonically decreasing timesteps during training, to reduce the randomness in timesteps.

**Guidance Calibration.** We further dive into the principle when LCM generates guidance. We first review that in DMs, the denoising process follows a reverse stochastic differential equation(SDE):

$$d\mathbf{x} = -\dot{\sigma}_t \sigma_t \nabla \log p_t(\mathbf{x}) dt + \sqrt{\dot{\sigma}_t \sigma_t}\, d\mathbf{w}, \quad (4)$$

where $p_t(x_t) \sim \mathcal{N}(x_0, \sigma_t^2 \mathbf{I})$, $\sigma_t$ varies along timestep $t$, $\dot{\sigma}_t$ is the time derivative of $\sigma_t$, $\mathbf{w}$ is the standard Wiener process and $\nabla \log p_t(\mathbf{x})$ is the score function which indicates the direction towards highest data density. And there exists a corresponding reverse ordinary deterministic equation(ODE) defined below:

$$d\mathbf{x} = -\dot{\sigma}_t \sigma_t \nabla \log p_t(\mathbf{x}) dt. \quad (5)$$

where $\nabla \log p_t(\mathbf{x})$ is estimated as $-\frac{1}{\sqrt{1-\bar{\alpha}_t}}\hat{\epsilon}_0^t$. LCM can map any $x_t$ on a trajectory of the PF-ODE to the origin $x_0^*$, which indicates the highest-data-density region. However, the mapped origin, i.e., $\hat{x}_0^t$ derived from single-step inference is always shifted. We attribute the shifting to the insufficient training of $f_\phi$. Our goal is to calibrate $\hat{x}_0^t$ to get closer to $x_0^*$. We consider the insufficient training in LCM and rationally assume that the denoising process of LCM follows a smooth PF-ODE trajectory with a small slope. This assumption allows us to calibrate the guidance from the perspective of PF-ODE.

Based on the analysis, we propose our guidance calibration strategy, which is a two-stage strategy. We repeat the perturbing and denoising steps to calibrate the guidance distribution. In the first stage, given a perturbed sample $x_s$ at timestep $s$, we first predict guidance $\hat{\epsilon}_\phi(x_s, y, s)$, where $s$ is set to a small number to make the guidance more consistent with $x_0$ than large $s$. In the second stage, since the denoising process of LCM follows PF-ODE, we run a discretization step of a numerical ODE solver. we use Euler Solver to get another sample $x_t$:

$$x_t = \frac{\sqrt{1-\sigma_s^2}x_s + (\sigma_t - \sigma_s)\hat{\epsilon}_\phi(x_s, y, s)}{\sqrt{1-\sigma_t^2}} \quad (6)$$

where $t > s$. We then fed $x_t$ to LCM network to obtain the final calibrated guidance $\hat{\epsilon}_\phi(x_t, y, t)$. Compared to $\hat{\epsilon}_\phi(x_s, y, s)$, the guidance $\hat{\epsilon}_\phi(x_t, y, t)$ has two advantages: 1) it is consistent with the original rendered image $x_0$ because the Euler solver makes $x_t$ and $x_s$ on the same PF-ODE trajectory; 2) large timestep $t$ provides more reasonable optimization direction conditioned on $y$, leading $\hat{x}_0^t$ closer to $x_0^*$. Overall, $\hat{x}_0^t$ is ensured to be consistent with both the rendered image and the PF-ODE trajectory origin conditioned on the text prompt, effectively improving the quality of the guidance. We optimize $\theta$ using the final guidance $\hat{x}_0^t$ to calculate SDS loss following Eq. (7) as

$$\nabla_\theta \mathcal{L}_{\text{SDS}}(\phi, g) = \mathbb{E}_{t,\epsilon,c}\left[w(t)\left(\hat{\epsilon}_\phi(x_t, y, t) - \epsilon\right)\frac{\partial g(\theta, c)}{\partial \theta}\right]. \quad (7)$$

**Dual Timestep Strategy.** In this paper, we incorporates 3D Gaussians [18] as the 3D representation, which requires initialization models, i.e., PointE [30], Shap-E [15] to initialize the geometry. However, these models sometimes initialize badly, especially when given complex text prompts. Thus, properly updating the geometry positions of the 3D model is crucial in 3D generation. We propose a two-phase strategy, optimizing the geometry and appearance of 3D objects in separate phases. In the initial phase, we use large timesteps to predict large deformations to the 3D model since large timesteps keep less information in the rendered image. As a result, the guidance includes global geometry features, leading DreamLCM to optimize the geometry, where the position of Gaussian Splatting is greatly updated. In the refinement phase, we use small timesteps to optimize the appearance because small timesteps keep more information on the rendered image, generating guidance with fine local features.

We propose a dual timestep strategy combining the decreasing timestep strategy with the two-phase strategy. Specifically, we define a cut-off iteration $T_{cut}$ and a cut-off timestep $t_{cut}$, in each

---

**Algorithm 1** DreamLCM

---

1: **Initialization**: 3D model parameters $\theta$, training iteration $n$, LCM network $\phi$ denoising timestep from $N_{min}$ to $N_{max}$, cut-off iteration $T_{cut}$ and timestep $t_{cut}$, text prompt $\boldsymbol{y}$, fixed noise $\epsilon'$.

2: **for** $i = [0, ..., n-1]$ **do**

3:    **if** $i \leq T_{cut}$ **then**

4:       $t_{max} \leftarrow N_{max}, t_{min} \leftarrow t_{cut}, t_{interval} \leftarrow T_{cut}, id \leftarrow i$

5:    **else**

6:       $t_{max} \leftarrow t_{cut}, t_{min} \leftarrow N_{min}, t_{interval} \leftarrow n - T_{cut},$ $id \leftarrow i - t_{cut}$

7:    **end if**

8:    **Sample**: *camera pose* $c, \boldsymbol{x}_0 = g(\theta, c)$

9:    $s \leftarrow t_{\max} - (t_{\max} - t_{\min}) \sqrt{id/t_{interval}}, t \leftarrow 2s$

10:    $\boldsymbol{x}_s \leftarrow \boldsymbol{x}_0 + \sigma_s \epsilon'$

11:    predict $\hat{\epsilon}_\phi(\boldsymbol{x}_s, \boldsymbol{y}, s)$

12:    **if** $i \leq T_{cut}$ **then**

13:       calculate SDS loss:

14:       $\nabla_\theta L_{\text{SDS}} = \omega(s) \left( \hat{\boldsymbol{\epsilon}}_\phi (\boldsymbol{x}_s, \boldsymbol{y}, s) - \epsilon \right)$, update $\theta$.

15:    **else**

16:       use Euler Solver to obtain $x_t$.

17:       predict $\hat{\epsilon}_\phi(\boldsymbol{x}_t, \boldsymbol{y}, t)$ then calculate SDS loss:

18:       $\nabla_\theta L_{\text{SDS}} = \omega(t) \left( \hat{\boldsymbol{\epsilon}}_\phi (\boldsymbol{x}_t, \boldsymbol{y}, t) - \epsilon \right)$, update $\theta$.

19:    **end if**

20: **end for**

---

stage, the timestep is calculated as follows :

$$t = t_{\max} - (t_{\max} - t_{\min}) \sqrt{T/N}, \quad (8)$$

where $T$ and $N$ are the current iteration and total iteration. For the first $T_{cut}$ iterations, we optimize geometry using timesteps larger than $t_{cut}$. For the remaining iterations, we use timesteps less than $T_{cut}$ to optimize appearance. Overall, we can see that the timestep strategy in HiFA [49] and ProlificDreamer [45] are two special cases of our timestep strategy. Experiments demonstrate that the strategy can generate high-quality 3D models with fine geometry and appearance, as shown in Fig. 5.

## 5 DISCUSSION

Similar to the proposed DreamLCM method, ProlificDreamer [45] and LucidDreamer [22] targets resolving the over-smooth issue in SDS. They refine the SDS loss with different loss functions to alleviate the over-smoothed and over-saturated results based on SDS. We will revisit these two losses to show the relationship between DreamLCM and the two works and demonstrate that our work is more effective than theirs.

**ProlificDreamer** is based on the SDS loss. It handles the over-smooth issue by training an additional LoRA [12] network denoted as $\epsilon_{LoRA}$. ProlificDreamer optimizes multiple 3D models simultaneously. It aggregates and estimates their distributions by finetuneing $\epsilon_{LoRA}$. The VSD loss for the $i_{th}$ 3D model is as follows:

$$\nabla_{\theta^{(i)}} \mathcal{L}_{\text{VSD}}(\theta^{(i)}) = \mathbb{E}_{t,\epsilon,c} \left[ \omega(t) \left( \hat{\epsilon}_\phi \left( \boldsymbol{x}_t^{(i)}, \boldsymbol{y}, t \right) - \hat{\epsilon}_{LoRA} \left( \boldsymbol{x}_t^{(i)}, \boldsymbol{y}, t, c \right) \right) \frac{\partial g(\theta^{(i)}, c)}{\partial \theta^{(i)}} \right], \quad (9)$$

where $\boldsymbol{x}_t^{(i)}$ is the rendered image of the $i_{th}$ 3D model and $c$ is the camera condition. $\hat{\epsilon}_{LoRA} \left( \boldsymbol{x}_t^{(i)}, \boldsymbol{y}, t, c \right)$ indicates the distribution of the rendered image. When ProlificDreamer optimizes one 3D model, the distribution of the rendered image can be estimated as $\epsilon$, where $\epsilon$ is the noise added to the rendered image. We consider the SDS gradient as the vector starting from $\epsilon$ and $\hat{\epsilon}_{LoRA}$ to $\hat{\epsilon}_\phi$. Since $\hat{\epsilon}_{LoRA}$ contains information from multiple 3D models, $\hat{\epsilon}_{LoRA}$ is a steadier and more robust starting point than $\epsilon$, averaging the random and inconsistent features in the optimization process of each 3D model.

We observe that the essential problem is the randomness and inconsistency when optimizing a single 3D model. Besides, $\hat{\epsilon}_{LoRA}$ introduces extra parameters and trains several 3D models simultaneously, leading to high training costs. However, the proposed DreamLCM method incorporates LCM as the guidance model, greatly mitigating the inconsistent issue when optimizing one 3D model. As a result, there is no need for DreamLCM to train another $\hat{\epsilon}_{LoRA}$ to decrease the training costs for generating high-quality 3D models.

**LucidDreamer** proposes ISM [22] loss, which employs DDIM Inversion to enhance the quality and consistency of the guidance. Specifically, it predicts a invertible noisy latent trajectory $\{\boldsymbol{x}_{\delta_T}, \boldsymbol{x}_{2\delta_T}, \ldots, \boldsymbol{x}_t\}$, iteratively following Eq. (10),

$$\boldsymbol{x}_t = \sqrt{\bar{\alpha}_t} \hat{\boldsymbol{x}}_0^s + \sqrt{1 - \bar{\alpha}_t} \boldsymbol{\epsilon}_\phi (\boldsymbol{x}_s, \emptyset, s), \quad (10)$$

where $s = t - \delta t$. The guidance is obtained by a multi-step DDIM denoising process i.e., iterating

$$\tilde{\boldsymbol{x}}_{t-\delta_T} = \sqrt{\bar{\alpha}_{t-\delta_T}} \left( \hat{\boldsymbol{x}}_0^t + \gamma (t - \delta_T) \boldsymbol{\epsilon}_\phi (\boldsymbol{x}_t, \boldsymbol{y}, t) \right), \quad (11)$$

where $\eta(t) = \frac{1 - \sqrt{\bar{\alpha}_t}}{\sqrt{\bar{\alpha}_t}}$. Next, by replacing $\hat{\boldsymbol{x}}_0^t$ in Eq. (2) with $\tilde{\boldsymbol{x}}_0$, the SDS loss can be rewrote as $\nabla_\theta \mathcal{L}(\theta) = \mathbb{E}_c \left[ \frac{\omega(t)}{\gamma(t)} \left( \boldsymbol{x}_0 - \tilde{\boldsymbol{x}}_0^t \right) \frac{\partial g(\theta, c)}{\partial \theta} \right]$. LucidDreamer then unifies the iterative process in Eq. (10) and Eq. (11), proposing ISM loss as follows:

$$\nabla_\theta \mathcal{L}(\theta) = \mathbb{E}_{t,c} \left[ \frac{\omega(t)}{\gamma(t)} \left( \gamma(t) \left[ \boldsymbol{\epsilon}_\phi (\boldsymbol{x}_t, \boldsymbol{y}, t) - \boldsymbol{\epsilon}_\phi (\boldsymbol{x}_s, \emptyset, s) \right] + \eta_t \right) \frac{\partial g(\theta, c)}{\partial \theta} \right]$$
$$\approx \mathbb{E}_{t,c} \left[ \omega(t) \left( \boldsymbol{\epsilon}_\phi (\boldsymbol{x}_t, \boldsymbol{y}, t) - \boldsymbol{\epsilon}_\phi (\boldsymbol{x}_s, \emptyset, s) \right) \frac{\partial g(\theta, c)}{\partial \theta} \right]. \quad (12)$$

where $\eta_t$ includes a series of neighboring interval scores with opposing scales, which can be disregarded. ISM essentially substitutes DDPM single-step inference [9] for DDIM multi-step inference [40] to generate high-quality and high-fidelity guidance $\tilde{\boldsymbol{x}}_0^t$.

However, the multi-step inference needs to forward the U-Net [36] in DMs multiple times, increasing training costs. Moreover, we can see that a key improvement in ISM is the quality and consistency of the guidance. Compared to LucidDreamer, DreamLCM is capable of generating high-quality and high-consistency guidance in a single-step inference. Consequently, DreamLCM is more effective with fewer training costs.

To sum up, we observe that the principal cause of the over-smooth issue in SDS is the inadequate quality of the guidance. These two methods tackle the issue by utilizing extra resources, e.g., training multiple NeRFs and DDIM Inversions, which is time-consuming. Differently, DreamLCM can resolve the over-smooth issue by taking full advantage of LCM, while saving training costs.

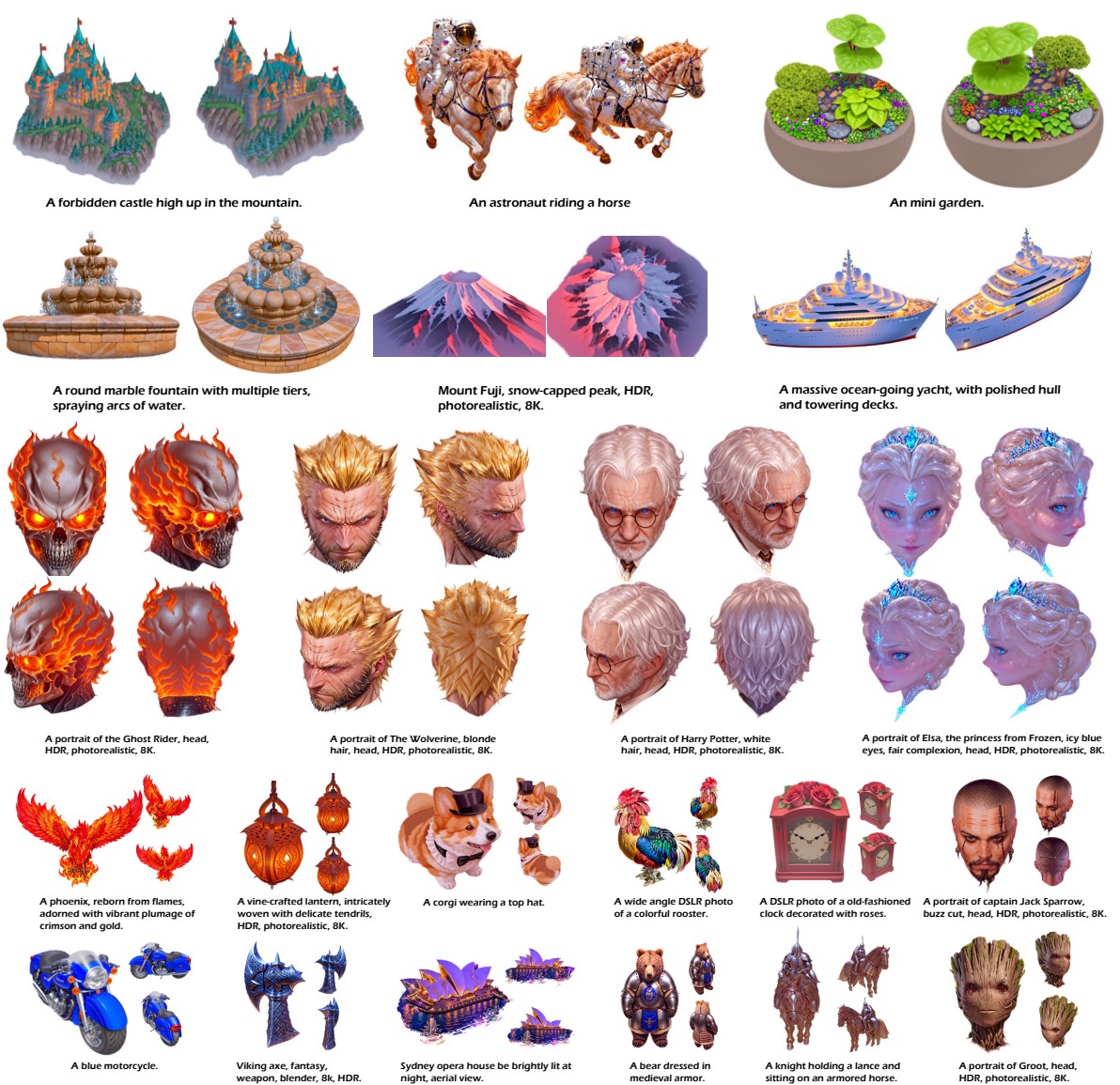

**Figure 3: Examples generated by DreamLCM. We incorporate the Latent Consistency Model (LCM) as a guidance model, with two proposed strategies to further enhance the generation quality (See section 4 for details). DreamLCM generates high-quality results with fine details.**

## 6 EXPERIMENTS

### 6.1 Implementation Details

We train our end-to-end network for 5000 iterations overall. We employ 3D Gaussian Splatting [18] as our 3D representation and 3D point cloud generation models Shap-E [15] and Point-E [30] for initialization. The rendering resolution is $512 \times 512$. As for the guidance calibration strategy, we use it in appearance optimization. We practically consider $s = 350$ as the cut-off timestep. Unless stated otherwise, we train the first 1000 iters for geometry optimization using timesteps $s$ fulfilling $350 \leq s \leq 980$ and the remaining 4000 iters for appearance optimization using timesteps $s$ fulfilling

$20 \leq s \leq 350$. Since we assume that LCM follows a smooth PF-ODE, the interval between $s$ and $t$ is less limited. Practically, we choose $t = 2s$. We use SDS with a normal CFG scale of 7.5. All experiments are performed and measured with an RTX 3090 (24G) GPU. We train about 50 min per sample.

### 6.2 Text-to-3D Generation.

In Fig. 4, we show the generated results of DreamLCM. We generate all examples using the original LCM without LoRA and any finetuned checkpoints. We can see that DreamLCM can generate photo-realistic 3D objects with fine details. The 3D objects are creative and highly consistent with the text prompts. Especially, we

| **DreamGaussian** | **GaussianDreamer** | **LucidDreamer** | **DreamLCM(ours)** |
|:---:|:---:|:---:|:---:|
| ( ∼ 2 mins) | ( ∼ 15 mins) | ( ∼ 1h) | ( ∼ 50 mins) |

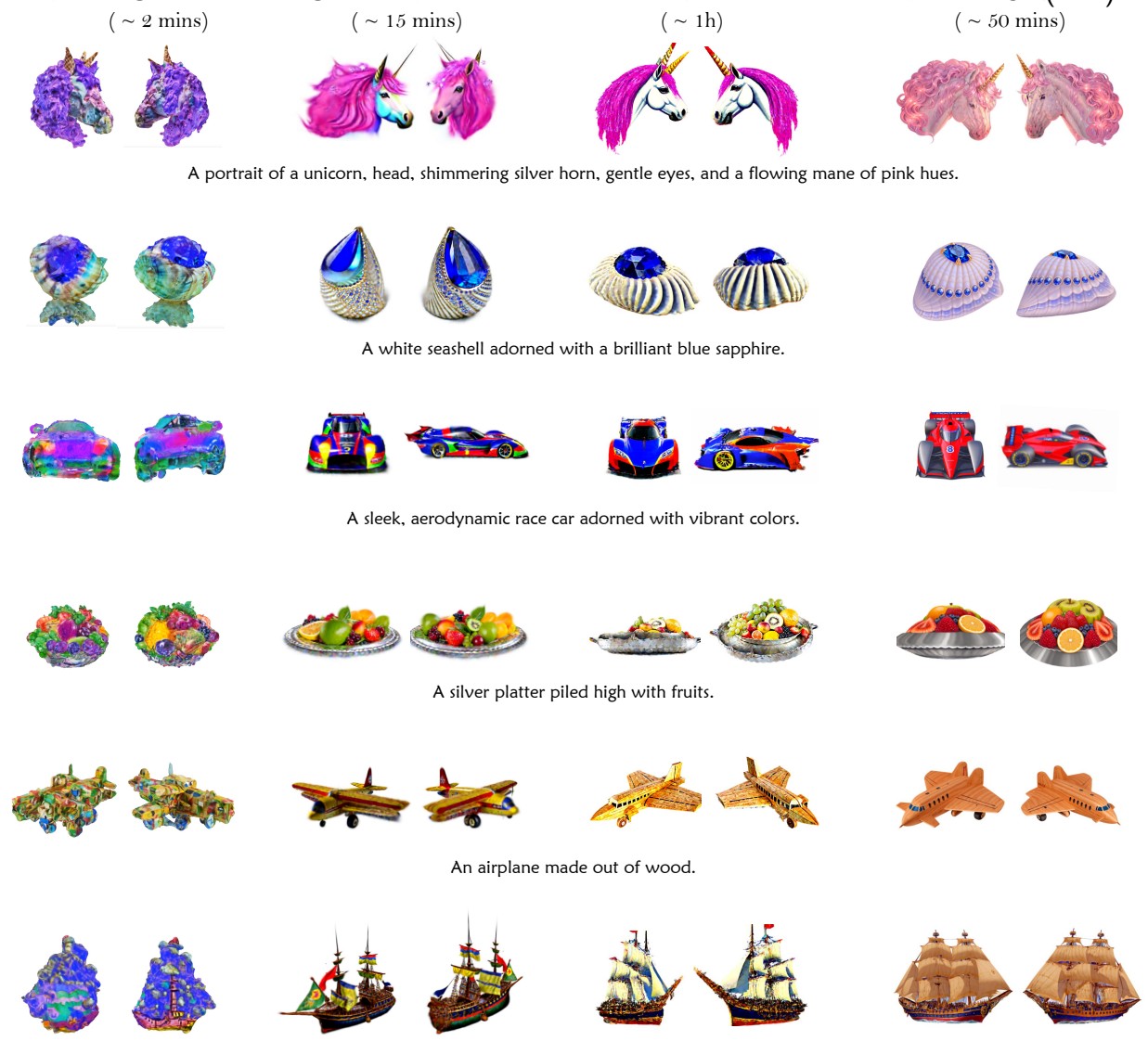

A portrait of a unicorn, head, shimmering silver horn, gentle eyes, and a flowing mane of pink hues.

A white seashell adorned with a brilliant blue sapphire.

A sleek, aerodynamic race car adorned with vibrant colors.

A silver platter piled high with fruits.

An airplane made out of wood.

A Spanish galleon.

**Figure 4: Comparison with the state-of-the-art text-to-3D generation methods with Gaussian Splatting as 3D representations. Experiments show that the proposed DreamLCM generates photo-realistic 3D objects with high quality and fine details. The models generated by DreamLCM are more consistent with the text prompt. The training time is measured with a single RTX 3090 GPU.**

can see that DreamLCM can generate different and amazing avatar heads conditioned on text prompts, such as *"A portrait of Harry Potter, white hair, head, HDR, photorealistic, 8K"*. Besides, the proposed DreamLCM method is good at generating objects conditioned on complex text prompts, like *"the fuji mountain"*, *"the massive yacht"*, and *"the fountain"*. These examples demonstrate that DreamLCM well resolves the over-smooth issue in SDS. Besides, these examples show great potential in generating all kinds of complex objects with different LCM finetuned checkpoints.

## 6.3 Qualitative Comparison

We compare our method with the current SoTA baselines which generate 3D Gaussian Objects [22, 43, 46]. As shown in Fig. 4, our model generates more photo-realistic results than other works, exhibiting high quality and fine details. For example, *"A portrait of a unicorn"* is more photo-realistic, and the fur is more silky than the results from the other three approaches. As for the results conditioned on the text *"A Spanish galleon"*, our model generates the most intact galleon, and the details of the hull are the finest

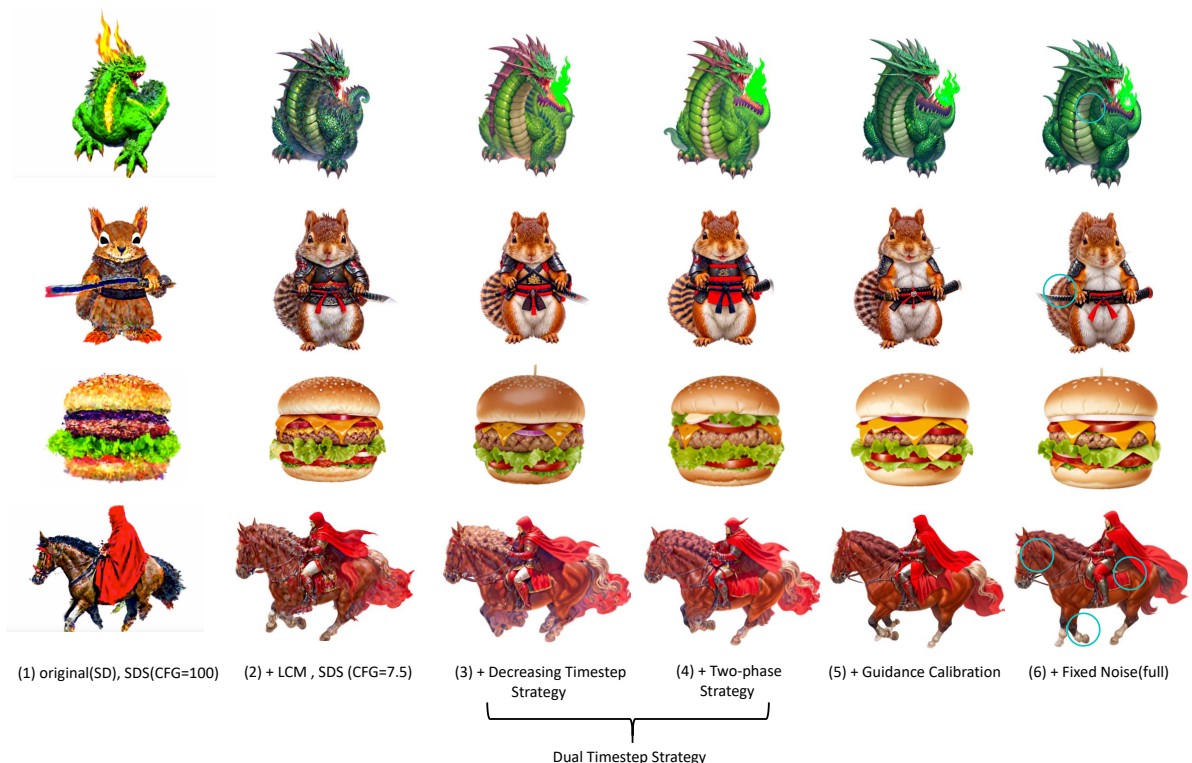

(1) original(SD), SDS(CFG=100)    (2) + LCM , SDS (CFG=7.5)    (3) + Decreasing Timestep Strategy    (4) + Two-phase Strategy    (5) + Guidance Calibration    (6) + Fixed Noise(full)

Dual Timestep Strategy

**Figure 5: Ablation Study of DreamLCM. The proposed components are effective and can improve the text-to-3D generation quality. (1) The results of SDS with a large CFG scale of 100. (2) We incorporate LCM as a guidance model with a small CFG of 7.5. (3)(4) The results after adding the Dual Timestep Strategy. It includes two parts, the Decreasing Timestep Strategy to reduce the randomness in timesteps and the Two-phases Strategy to improve geometry. Both parts are effective. (5) The results after adding Guidance Calibration to further improve the generation quality. (6) We use fixed noise to perturb the samples to reduce the randomness in noises to improve the details. We highlight some improved details in cyan. The prompts corresponding to the four examples are _"a green dragon breathing fire"_, _"a squirrel in samurai armor wielding a katana"_, _"a delicious hamburger"_ and _"A warrior with red cape riding a horse"_.**

among all the shown methods. Notably, compared to LucidDreamer, we generate higher quality objects with less training costs.

## 6.4 Ablation Study

Fig. 5 depicts the ablation experiments of different baseline methods. In Fig. 5(b) and (c), we utilize timesteps between 20 and 500 to generate high-quality images. Other settings are the same as the final settings 6.1. Starting from the original SDS loss, guided by Stable Diffusion [35], with a large CFG scale(100). We first incorporate LCM as the guidance model to demonstrate that LCM is a superior guidance model to DMs [35]. We can see that LCM makes a huge improvement in generation quality. We then add our Dual Timestep Strategy. We divide the strategy into two parts. We demonstrate the effectiveness of Decreasing Time Strategy and the Two-phase Strategy, as shown in Fig. 5(c) and (d). We can see that the hamburger adding Decreasing Time strategy shows a quality improvement. Based on (c), the hamburger adding the two-phase strategy shows a geometric advancement of the bread at the bottom of the hamburger. Besides, due to the two-phase strategy, the worrier example is deformed to be less close to the horse, since

this 3D model is initialized by the prompt _"a wolf"_. Then, we add the guidance calibration strategy which smooths the appearance and improves the details, making the objects more photo-realistic. Finally, we add fixed noises to improve the consistency of guidance between different timesteps. As shown in Fig. 5, the ability to improve details is demonstrated in cyan, such as the eye, legs, and cushion on the horse back in the worrier sample, the katana in the squirrel sample and the shadow in the dragon sample.

## 7 CONCLUSION

In this paper, we propose DreamLCM method to improve the text-to-3D object task. We incorporate LCM as a guidance model to generate high-quality guidance to resolve the two factors that cause the over-smooth issue. Besides, we introduce two techniques, i.e., Guidance Calibration and Dual Timestep Strategy, to further improve the generation quality. Experiments show superior performance of our method. Our method achieves state-of-the-art results in both generation and training efficiency.

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

Received 20 February 2007; revised 12 March 2009; accepted 5 June 2009