# OpenReview forum: "DreamLCM: Towards High Quality Text-to-3D Generation Via Latent Consistency Model"
_acmmm.org/ACMMM/2024/Conference — MM2024 Poster_

### Official Review · Reviewer_BvNT · 2024-05-04

**Rating:** 5
**Confidence:** 4

**Summary:**

This paper presents a training strategy for high-quality text-to-3D generation. The introduction of the Latent Consistency Model (LCM) can improve training stability. Guidance calibration has also been proven to be useful. Keeping the noise added in the rendered images constant is also an interesting and useful trick. The experiments show that the whole training pipeline produces impressive results.

**Strengths:**

1. This paper is well-written, which makes readers understand the claims and method easily.
2. DreamLCM produces impressive generation quality.
3. Many tricks are introduced into the training pipeline to ensure the generation quality.
4. The introduction of LCM is useful, and the guidance calibration is also an effective patch for the LCM in the text-to-3D pipeline.
5. The ablation studies clearly show the effectiveness of different parts of the pipeline.

**Limitations:**

1. This paper ignores an important reference, DreamTime [1]. DreamTime has proposed the monotonically non-increasing functions to control the timestep t. This is significantly similar to one of the tricks used in the DreamLCM. However, there is a lack of citations for this paper.
2. In Line 143, the authors mentioned that one of the merits is to keep the original SDS. And they mainly illustrate this point from the complexity of the loss function. The logic looks strange.


References

[1] Huang Y, Wang J, Shi Y, et al. DreamTime: An Improved Optimization Strategy for Diffusion-Guided 3D Generation[C]//The Twelfth International Conference on Learning Representations. 2023.

**Suitability:**

3

---

### Official Review · Reviewer_2PRm · 2024-05-23

**Rating:** 5
**Confidence:** 3

**Summary:**

This paper proposes a new text-to-3D framework, DreamLCM, which combines Latent Consistency Model (LCM) with its powerful image generation ability to generate consistent and high-quality 3D results, avoiding the problem of over-smoothing which leads to quality imbalance in SDS loss. In addition, two strategies are proposed to further improve the generation quality: one is a guided calibration strategy, using the Euler solver to calibrate the guided distribution, and the other is a dual time-step strategy, optimizing the 3D model from geometry to appearance

**Strengths:**

1. **An effective combination with LCM model instead of conventional diffusion**: It is effective to use LCM instead of the diffusion model to reduce the excessive smoothing problem caused by SDS loss, and improve the quality and fineness of the generated model.

2. **Clear writing with explainable figures**: The writing and figures are clear to understand the motivation and implementation of the work. Also, it provides an insightful discussion with ProlificDreamer and LucidDreamer, showing clearer intuition.

3. **High-fidelity Results**: The experiments show high-quality results in text-to-3D generation without Janus Problem.

**Limitations:**

1. **geometric inconsistencies**: Note that in the model generated by the ablation experiment, the horse appears to have only three legs, which miraculously disappear after the addition of certain components (such as the time-step reduction strategy), but reappear after the addition of the two-step strategy. This seems to be the case with the horse in Figure 3.

2. **Lack of baseline for comparison**: Now there are many text-to-3D models, many of which are dedicated to solving the problem of excessive SDS smoothness, such as ProlificDreamer mentioned in the article, you can compare with such models to generate results, operation time, cost and so on. Rather than a model like DreamGaussian that addresses the build speed issue. In addition, it is not necessary to limit the model generated by the Gaussian splatting method.

3. **Some ablation experiments are missing**: The ablation experiment lacks a model with guidance calibration and no two-time-step strategy. In addition, it may be possible to add background information and formulas on SDS losses.

4. **More discussion**: Choosing a more advanced model (e.g., MVDream), which is a multi-view aware diffusion at the start of LCM may be a more reasonable approach. But it is a minor problem.

**Suitability:**

3

---

### Official Review · Reviewer_hFVm · 2024-05-24

**Rating:** 4
**Confidence:** 3

**Summary:**

This paper introduces LCM into the optimize-based 3D generation pipline to address the over-smooth problem. Firstly, in order to improve the quality of guidance during refinement, this paper proposes guidance calibration module, which aligns the noisy latents and reference image latents on the same PF-ODE trajectory. Secondly, this paper proposes dual time strategy, which optimizes geometry and appearance with large and small timesteps respectively when decreasing the timestep at the same time.

**Strengths:**

1.This paper introduce LCM into optimize-based 3D generation pipeline for the first time.

2.The proposed guidance calibration are well-motivated and effective.

3.The experiment results on 3D generation are convincing.

**Limitations:**

1.Further analysis. In guidance calibration module, authors demostrate that the shifting of images is caused by the insufficient training and assume that the denoising process of LCM follows a smooth PF-ODE trajectory with a small slope. Could authors provide further theoretical analysis on the relationship between the insufficient training and the smooth PF-ODE trajectory?

2.Effectiveness of dual time strategy. As shown in Fig.5, the results w/o dual time strategy(column2) actually have more geometry details than the results with the strategy(column3&4), such as the tail of the dragon and the front leg of the horse. Perhaps the authors need to further demonstrate the effectiveness of the strategy with experiment results.

3.Necessity of two-phase strategy. As far as I'm concerned, NeRF is more stable than gaussian splatting during the optimizing process. Is the two-phase strategy still necessary when training on NeRF, as the decreasing timestep strategy is widely used in previous work?

**Suitability:**

3

---

### Official Review · Reviewer_SqFZ · 2024-05-26

**Rating:** 3
**Confidence:** 3

**Summary:**

This paper is the first to apply the CM method to SDS, introducing a novel approach. The authors claim that this method resolves the over-smoothing problem.

**Strengths:**

1) This paper leverages the LCM model to incorporate more information, resulting in improved generation.

2) The paper is well-written and clearly articulated.

**Limitations:**

1) The application of LCM is quite strange. Firstly, the SDS loss originates from the training loss of diffusion models, which involves taking the derivative with respect to the image to obtain the SDS loss. Directly switching the model from a diffusion model to an LCM model is strange because the LCM loss does not follow this approach. Secondly, for LCM, it maintains the trace, but  $x_0$  obtaining  $x_s$  does not follow the ODE, which means the predicted target of LCM from $x_s $ is not $x_0 $, resulting in inconsistency. Lastly, to perform Euler Solver, diffusion models are still needed, leading to double memory requirements.

2) Algorithm 1, $ L_{SDS} $, lacks a term.

3) The results do not appear to be better than those of LucidDreamer.

If you can address these concerns, I will raise my score.

**Suitability:**

3

---

### Meta-Review · Area_Chair_3AeU · 2024-06-27

**Recommendation:** Accept (Poster)
**Confidence:** 5

**Metareview:**

This paper introduces LCM into optimize-based 3D generation pipeline, which is novel and effective. All the reviewers agree to accept the manuscript. I recommend a decision of acceptance. In the final version, the authors should include the discussions in the rebuttal to address the concerns of reviewers.